

# Complementary Relationship for Estimating Evapotranspiration Using the Granger-Gray Model: Improvements and Comparison with a Remote Sensing Method

Homin Kim[1], Jagath J. Kaluarachchi[2]

[1]Utah Water Research Laboratory, Utah State University, 1600 Canyon Road, Logan, UT 84321, USA
[2]College of Engineering, Utah State University, 4100 Old Main Hill, Logan, UT 84322-4100, USA

*Correspondence to*: Homin Kim (homin.kim@aggiemail.usu.edu)

**Abstract.** The Granger and Gray (GG) model, which uses the complementary relationship for estimating evapotranspiration (ET), is a simple approach requiring only commonly available meteorological data; however, most complementary
relationship models decrease in predictive power with increasing aridity. In this study, a previously developed modified GG model using the vegetation index is further improved to estimate ET under a variety of climatic conditions. This updated GG model, GG-NDVI, includes Normalized Difference Vegetation Index (NDVI), precipitation, and potential evapotranspiration using the Budyko framework. The Budyko framework is consistent with the complementary relationship and performs well under dry conditions. We validated the GG-NDVI model under operational conditions with the commonly used remote
sensing-based Operational Simplified Surface Energy Balance (SSEBop) model at 60 Eddy Covariance AmeriFlux sites located in the USA. Results showed that the Root Mean Square Error (RMSE) for GG-NDVI ranged between 15 and 20 mm month$^{-1}$, which is lower than for SSEBop every year. Although the magnitude of agreement seems to vary from site to site and from season to season, the occurrences of RMSE less than 20 mm month$^{-1}$ with the proposed model are more frequent than with SSEBop in both dry and wet sites. This study also found an inherent limitation of the complementary relationship
under moist conditions, indicating the relationship is not symmetrical as previously suggested. A nonlinear correction function was incorporated into GG-NDVI to overcome this limitation. The resulting Adjusted GG-NDVI produced much lower RMSE values, along with lower RMSE across more sites, as compared to measured ET and SSEBop.

## 1 Introduction

According to the U.S. Geological Survey (USGS) Famine Early Warning Systems NETwork (FEWSNET, 2015), the rate
and amount of evapotranspiration (ET) plays a considerable role in the monitoring of water loss from agricultural lands. As noted by Senay et al. (2013), ET may be used to show the current vegetation condition compared to the historical records. This comparison has the potential to help identify vegetation stress in time and space. ET estimation methods can be divided into two types: (1) ground-based ET methods that use standard meteorological data; and (2) ET models that use remote sensing data that must be combined with retrieval algorithms to estimate ET.



McMahon et al. (2016) classified the ground-based ET methods into six classes on the basis of application: (1) potential evapotranspiration (ETP); (2) reference evapotranspiration; (3) actual evapotranspiration; (4) open water evaporation; (5) lake/storage evaporation; and (6) pan evaporation. We have focused on actual ET in this study because it can be representative of actual conditions, whereas reference evapotranspiration would require a vegetation resistance parameter and deep lakes would require water temperature data. In addition, we use the term 'evapotranspiration (ET)' in this paper to include actual evapotranspiration except in places where the term 'reference (crop) evapotranspiration' is used by other authors.

One approach to estimating ET with ground-based methods is the complementary relationship proposed by Bouchet (1963). The primary advantage of the complementary relationship is that it generally requires only meteorological data. Bouchet (1963) suggested that as a surface dries, the decrease in ET is matched with an increase in potential evapotranspiration (ETP). Such a relationship offers a simple and attractive approach for estimating ET using ETP without the detailed knowledge of surface properties. Examples of widely known models using this concept are the Advection-Aridity (AA) model by Brutsaert and Stricker (1979), the Complementary Relationship Areal Evapotranspiration (CRAE) by Morton (1983), and the GG model proposed by Granger and Gray (1989). These three models have been widely applied to a broad range of surface and atmospheric conditions (Hobbins et al., 2001; Kahler and Brutsaert, 2006; Szilagyi and Jozsa, 2008; Xu and Singh, 2005).

Granger (1989), however, argued that the symmetric relationship in Bouchet (1963) lacked a theoretical background and proved that the symmetric condition is only true when the temperature is near 6 ˚C. Hence, the author developed a new complementary relationship with the psychrometric constant and the slope of the saturation vapor pressure curve. Later, Cargo and Crowley (2005) showed that the radiometric surface temperature measurements can be successfully incorporated into the Granger (1989) equation. Similar to Cargo and Crowley (2005), Anayah and Kaluarachchi (2014) proposed a modified version of the GG model using the Priestley and Taylor (1972) equation instead of the Penman (1948) equation. The model proposed by Anayah and Kaluarachchi (2014) is hereafter called the modified GG model. The results of the modified GG model showed a decrease in Root Mean Square Error (RMSE) from 20 % to as much as 80 % compared to the recent studies of Mu et al. (2007, 2011), Szilagyi and Kovacs (2010), Han et al. (2011), and Thompson et al. (2011). On the other hand, Kahler and Brutsaert (2006) proposed an empirical constant, b, in the Bouchet (1963) hypothesis and demonstrated that b is generally greater than 1, based on their theoretical and experimental evidence, while the symmetric condition of the Bouchet (1963) hypothesis requires b = 1. More recently, Aminzadeh et al. (2016) extended the asymmetric complementary relationship with an analytical prediction of b for Kahler and Brutsaert (2006). Furthermore, Venturini et al. (2008; 2011) applied surface temperature of Moderate Resolution Imaging Spectroradiometer (MODIS) data into the GG model and showed a good agreement between their approach and measured ET.

Prior studies show that the complementary relationship is not symmetric with ETW and that the GG model can be successfully applied to a wide range of physical and surface conditions. Specially, the modified GG model (Anayah and Kaluarachchi, 2014) provided more reliable ET estimates than other models. Although the modified GG model demonstrated





excellent performance across 34 global sites, the authors suggested that additional refinements could further improve performance under dry conditions. The low performance in dry conditions may be due to relative evaporation (the ratio of ET to ETP) in the original GG model (Granger and Gray, 1989), which was empirically derived from 158 sites under wet conditions in Canada. Therefore, models based on the original GG may have difficulty predicting ET under dry conditions.

To improve relative evaporation, Kim and Kaluarachchi (2017) used the Budyko model equation described by Li et al., (2013) to represent relative evaporation instead of using the original equation. The basis for this change is that the concept of relative evaporation is consistent and similar to that described in the Budyko framework (Yang et al., 2006; Zhang et al., 2004). Kim and Kaluarachchi (2017) selected 75 Eddy Covariance (EC) flux tower sites across the USA and compared them with measured ET and with other complementary relationship models. The Kim and Kaluarachchi (2017) model reduced

mean RMSE by 32 % compared to the Anayah and Kaluarachchi (2014) modified GG model across 36 dry sites. Using the Kim and Kaluarachchi (2017) model, the lowest mean RMSE across the 59 sites was shown to be 14 mm month-1, compared to 21 mm month-1 with CRAE, 28 mm month-1 with AA, 27 mm month-1 with GG, and 17 mm month-1 with the modified GG model.  Moreover, the predicted ET values were more correlated with estimated ET, showing a correlation coefficient of 60 % compared to 37 % in the Allam et al. (2016) study.

Figure 1 presents the results obtained from the previous Kim and Kaluarachchi (2017) study. These results are in agreement with Anayah and Kaluarachchi (2014), which showed that the modified GG model needs further improvements in dry conditions, and showed the lowest mean RMSE in both dry and wet sites. Overall, these results indicate that, among the ground-based methods, the Kim and Kaluarachchi (2017) model can be used as a powerful methodology to estimate ET.

While these findings are good within the realm of complimentary methods (or ground-based methods), some of the more

commonly used ET estimation methods now use remote sensing data. If the complementary relationship and the corresponding methods, such as the model proposed by Kim and Kaluarachchi (2017), are to be accepted as operational models in field conditions, then the results should be compared and validated with remote sensing-based ET estimation methods. Taking into consideration of the improvements made with complementary relationship-based methods, this study examines the work of Kim and Kaluarachchi (2017) in comparison with a commonly used remote sensing method and

measured ET data from 60 EC flux tower sites located across the USA.

Biggs et al. (2016) grouped the remote sensing-based methods into three classes: vegetation-based methods, radiometric land surface temperature-based methods, and triangle/trapezoid or scatterplot inversion methods. Among them, the radiometric land surface temperature-based methods have a number of attractive features compared to the other classes: minimal ground data, ease of implementation, and operational application over large areas.

Radiometric land surface temperature-based methods use the fact that ET is a change of state in water that uses energy in the environment for vaporization and reduces surface temperature (Su et al., 2005). A subset of these methods is often called energy balance methods since they solve the energy balance equation. Moreover, these methods do not directly measure ET but must be combined with retrieval algorithms since data and technical requirements to solve the full energy balance equation can be challenging, especially in large regions. For example, the Surface Energy Balance Algorithm for Land



(SEBAL) model (Bastiaanssen et al., 1998; 2005) requires the measurements of wind speed, iterative calibration, and review by an expert operator. Mapping EvapoTranspiration at high Resolution with Internalized Calibration (METRIC) (Allen et al., 2011) needs high-quality meteorological data such as net radiation, air temperature, wind speed, and humidity. According to Allen et al. (2011), METRIC has higher accuracy for hourly reference ET than SEBAL, but the processing cost of METRIC

is high.

As an alternative, FWESNET (USGS) has produced ET measurements from MODIS using the operational Simplified Surface Energy Balance (SSEBop) model (Senay et al., 2013). The SSEBop setup uses the Simplified Surface Energy Balance (SSEB) approach developed by Senay et al. (2007). The SSEB approach estimates ET using ET fraction scaled from thermal imagery in combination with a spatially explicit maximum reference ET. SSEB has an advantage in that it does not

require air temperature and the knowledge of land cover types. Instead, the method uses the 'hot' and 'cold' pixel approach of Bastiaanssen et al. (1998) to calculate the ET fraction. Gowda et al. (2009) found a strong correlation of 0.84 between SSEB results and lysimeter data. Later, Senay et al. (2011a) enhanced SSEB to accommodate diverse vegetation and topographic conditions using a lapse rate correction factor. They successfully evaluated the results by comparing with METRIC and ET values computed from the water balance approach. As a result of the work by Senay et al. (2011a), the

enhanced SSEB model increased the correlation with METRIC from 0.83 to 0.90. Furthermore, Senay et al. (2011b) proposed a revised SSEB to handle both elevation and latitude effects on surface temperature using the difference between Land Surface Temperature (LST) and air temperature. Recently, Senay et al. (2013) proposed an operational SSEB, renamed as SSEBop, that uses predefined boundary conditions for hot and cold reference pixels so that ET can be calculated as a function of LST and reference ET. The SSEBop approach has been validated comprehensively by comparing with 45 EC

flux tower observations (Senay et al., 2013) and then with both MOD16 and 60 EC flux tower observations (Velpuri et al., 2013). Later, Bastiaanssen et al. (2014) applied SSEBop to determine ET in the Nile Basin, Ethiopia, for mapping water production and consumption zones. SSEBop ET data is now freely available through the USGS Geo Data Portal.

Despite the general consensus of using SSEBop for estimating ET, a detailed study of SSEBop conducted by Senay et al. (2013) showed that the use of reference ET can introduce a significant difference of up to 20 % in the magnitude of ET.

They also showed that the use of constant pre-defined differential temperature between the hot and cold boundary conditions can also create an inherent inaccuracy. Thus, it is important that SSEBop ET be validated and calibrated with available data such as EC flux tower data before using it to model ET.

The facts provided in the previous discussion indicate a need to further validate both the Kim and Kaluarachchi (2017) and SSEBop models in the operational application of the complementary relationship in estimating ET. Therefore, the objectives

of this study are: (1) assess the validity of the ET estimation model of Kim and Kaluarachchi (2017) through a direct comparison with remote sensing methodology, which in this case is the SSEBop model; and (2) use the results of the first objective to identify the potential improvements required in the complementary relationship for estimating ET under diverse climate conditions.



## 2 Methodology and Data

### 2.1 Methodology

GG-NDVI is the most updated model using the original GG model. GG-NDVI uses historical annual Normalized Difference Vegetation Index (NDVI) data and precipitation to improve the ET estimates of the modified GG model proposed by Anayah

and Kaluarachchi (2014). We then used the SSEBop model (Senay et al., 2013) to further validate GG-NDVI in comparison to an operational remote sensing model.

### 2.1.1 GG-NDVI model

The first complementary relationship was proposed by Bouchet (1963), who postulated that, as a surface dries, the actual ET decrease is matched by an equivalent increase in ETP. In spite of the fact that ET is negatively correlated with ETP, Morton

(1983) showed that the relationship has no defined shape. Granger (1989) showed that the symmetrical relationship between ET and ETP only occurs when the temperature is near 6 ˚C and suggested the following complementary relationship formulation:

$$\text{ET} + \frac{\gamma}{\Delta}\text{ETP} = \left(1 + \frac{\gamma}{\Delta}\right)\text{ETW} \tag{1}$$

where ET, ETP, and ETW are in mm day$^{-1}$, $\gamma$ is the psychrometric constant (kPa ˚C$^{-1}$), and $\Delta$ is the slope of saturation vapor

pressure-temperature (kPa ˚C$^{-1}$) relationship. Thereafter, Granger and Gray (1989) developed the GG model based on Eq. (1) using the concept of relative evaporation. Recently, Anayah and Kaluarachchi (2014) developed the modified GG model using the work of Granger and Gray (1989). The performance of the modified GG model improves when the Priestley and Taylor (1972) equation shown in Eq. (2) is used to calculate ETW instead of the Penman (1948) model.

$$\text{ETW} = \alpha \frac{\Delta}{\gamma+\Delta}(R_n - G_{soil}) \tag{2}$$

where $\alpha$ is a coefficient equal to 1.28, $R_n$ is net radiation (mm day$^{-1}$), and $G_{soil}$ is soil heat flux density (mm day$^{-1}$). Note that soil heat flux density is negligible compared to net radiation when calculated at daily or monthly time-scale (Gavilana et al., 2007; Hobbins et al., 2001).

ET is then estimated as a fraction of ETW using Eq. (3):

$$\text{ET} = \frac{2G}{G+1}\text{ETW} \tag{3}$$

where $G$ is the relative evaporation parameter derived from Granger and Gray (1989). They proposed a unique relationship with a parameter called relative drying power ($D$). The unique relationship between $G$ and $D$ are described in Eqs. (4) and (5), respectively.

$$G = \frac{\text{ET}}{\text{ETP}} = \frac{1}{1+0.028e^{8.045D}} \tag{4}$$

$$D = \frac{E_a}{E_a+R_n} \tag{5}$$

where $E_a$ is drying power of air (mm day$^{-1}$) given in Eq. (6).



$$E_a = 0.35(1 + 0.54U)(e_s - e_a) \tag{6}$$

where $U$ is wind speed at 2 m above ground level (m s$^{-1}$), which is adjusted using the work of Allen et al. (1998); $e_s$ is saturation vapor pressure (mm Hg); and $e_a$ is vapor pressure of air (mm Hg).

The performance of the GG model, including the modified GG model proposed later, decreased with increasing aridity. A possible reason is $G$ in Eq. (4), which was empirically derived from 158 sites representing wet environments in Canada. To improve the parameter $G$, the Kim and Kaluarachchi (2017) GG-NDVI model used the latest version of the Fu equation (Li et al., 2013). In particular, the Fu (1981) equation is one of the formulations of the Budyko curve (Budyko, 1974) and it is consistent with the complementary relationship (Yang et al., 2006; Zhang et al., 2004). The corresponding analytical formulation of the Fu equation is given in Eq. (7).

$$\frac{ET}{ETP} = 1 + \frac{P}{ETP} - \left[1 + \left(\frac{P}{ETP}\right)^{\varpi}\right]^{\frac{1}{\varpi}} \tag{7}$$

where $P$ is precipitation (mm) and ETP is estimated using Penman (1948). Parameter $\varpi$ is a constant and represents the land surface conditions of the basin, especially the vegetation cover (Li et al., 2013). Furthermore, Li et al. (2013) showed that $\varpi$ is linearly correlated with the long-term average annual vegetation cover that can help improve ET estimates. Yang et al. (2009) showed that vegetation cover defined by $M$ is calculated using Eq. (8).

$$M = \frac{NDVI - NDVI_{min}}{NDVI_{max} - NDVI_{min}} \tag{8}$$

where NDVI$_{min}$ and NDVI$_{max}$ are chosen to be 0.05 and 0.8, respectively. An optimal $\varpi$ value for the basin can be derived through a curve fitting procedure that minimizes RMSE between the measured and predicted evaporation ratio (Li et al., 2013).

Li et al. (2013) proposed parameterization that is simply a linear regression between optimal $\varpi$ and the long-term average $M$ given as

$$\varpi = a \times M + b \tag{9}$$

where $a$ and $b$ are constants that are found for each site.

To incorporate Eq. (7) into the modified GG model, Kim and Kaluarachchi (2017) used the work of Zhang et al. (2004) and Yang et al. (2006). According to Zhang et al. (2004), the Fu equation showed that the rate of change of ET with precipitation increases with ETP but decreases with precipitation. This is similar to the complementary relationship proposed by Bouchet (1963). Later, Yang et al. (2006) derived the consistency between the Fu equation and the complementary relationship using 108 dry regions in China. With this theoretical background, Kim and Kaluarachchi (2017) used the Fu equation to calculate $G$ in the modified GG model instead of Eq. (4). Equation (10) shows the Fu equation with the updated $G$ now defined as $G_{new}$.

$$G_{new} = \frac{ET}{ETP} = 1 + \frac{P}{ETP} - \left[1 + \left(\frac{P}{ETP}\right)^{\varpi}\right]^{\frac{1}{\varpi}} \tag{10}$$




Note $G_{new}$ is the updated definition of relative evaporation, $G$, which includes the Budyko hypothesis and the vegetation index. To estimate $G_{new}$, ETP is required and can be estimated using the Penman equation given by Eq. (11).

$$\text{ETP} = \frac{\Delta}{\gamma + \Delta}(R_n - G_{soil}) + \frac{\gamma}{\gamma + \Delta}E_a \tag{11}$$

Having found $G_{new}$ from Eq. (11) and estimated ETW from Eq. (2), we can estimate ET of the proposed model from Eq. (12).

$$\text{ET} = \frac{2G_{new}}{G_{new} + 1}\text{ETW} \tag{12}$$

### 2.1.2 SSEBop Model

The SSEBop algorithm (Senay et al., 2013) does not solve the full energy balance equation. This approach assumes that for a given time and location, the temperature difference between the hot and cold reference values of each pixel remains nearly constant throughout the year under clear sky conditions. Furthermore, the major simplification of SSEBop is based on the knowledge that the surface energy balance process is mostly driven by net radiation. With this simplification, the ET fraction, $ETf$, is calculated using Eq. (13).

$$ETf = \frac{Th - Ts}{dT} = \frac{Th - Ts}{Th - Tc} \tag{13}$$

Here, $ETf$ is between 0 and 1, with negative $ETf$ values set to zero; $Ts$ is surface temperature derived from MODIS LST; $Th$ is hot reference value representing the temperature of hot conditions; and $Tc$ is the cold reference value derived as a fraction of maximum air temperature (Senay et al., 2013). The difference between $Th$ and $Tc$ is $dT$ with temperature units in Kelvin.

ET is estimated using Eq. (14) as a fraction of reference ET.

$$\text{ET} = ETf \times k\, ET_o \tag{14}$$

where $ET_o$ is reference ET, which is calculated from the Penman-Monteith equation (Allen et al., 2007; Senay et al., 2008), and $k$ is a coefficient that scales $ET_o$ into the level of maximum ET experienced by an aerodynamically rougher crop. A recommended value of $k$ for the United States is 1.2, and the actual magnitude of $k$ should be determined using a validation or calibration process with field data.

### 2.2 Data

First, we used the SSEBop ET data set from the USGS Geo Data Portal (http://cida.usgs.gov/gdp/, last accessed on May 23, 2016) for the period 2000–2007 covering the United States. Second, ET data from GG-NDVI were generated using meteorological data and NDVI. Meteorological data required are temperature, wind speed, precipitation, net radiation, and elevation (pressure). Among these, net radiation ($R_n$) was calculated using the equations recommended by Allen et al. (2007), similar to the SSEBop model. Air temperature, elevation, and precipitation data were obtained from the Parameter-




elevation Regressions on Independent Slopes Model (PRISM) (http://www.prism.oregonstate.edu/, last accessed on Nov 23, 2015). As part of the input data for the GG-NDVI method, we used the 16-day Normalized Difference Vegetation Index (NDVI) data from MODIS (http://daac.ornl.gov/MODIS/modis.shtml, last accessed on Oct 23, 2015).

We used EC flux tower data (in mm month$^{-1}$) from 60 AmeriFlux stations for validation of ET results from SSEBop and GG-NDVI ET products. The details related to these sites are shown in Fig. 2. We used level 4 latent heat flux data from 60 sites for years 2000–2007; these data were collected from Oak Ridge Nation Laboratory's AmeriFlux website (http://ameriflux.ornl.gov/, last accessed on Nov 23, 2015). Level 4 data is gap-filled and quality-checked and does not require filling of the missing data. The measured monthly latent heat flux data were used to calculate the corresponding ET using latent heat of vaporization of water.

We defined the climate class of each site using the aridity index of the United Nations Environment Programme (UNEP) proposed by Barrow (1992). The aridity index divided climate conditions to six classes: hyper-arid, arid, semi-arid, dry sub-humid, wet sub-humid, and humid. However, this work simplified the climate class definition to two classes, similar to the work of Anayah and Kaluarachchi (2014): dry and wet. Using this simplification, 24 sites were identified as dry, compared to 36 sites under the wet class.

## 3 Results and Discussion

This study was conducted in two phases. Phase 1 is the validation stage in which comparisons are made between the SSEBop model and measured ET to assess the accuracy of the remote sensing method to estimate ET. In Phase 2, a comparison of estimated ET from GG-NDVI with observed data will be performed to identify the weaknesses of the GG-NDVI model, especially relative to the complementary relationship, and appropriate corrections will be proposed.

### 3.1 Phase 1: Validation of GG-NDVI

Capturing inter-annual variations of ET estimates is important. Although such variations are not significant when water is unlimited, estimating these variations in water-limited conditions is essential for water resources management. In this phase, ET has been estimated from both SSEBop and GG-NDVI and compared against measured monthly ET data from 2000 to 2007.

Table 1 presents the yearly comparison of results between the SSEBop and GG-NDVI estimates. Compared with measured ET, the results indicate that the accuracy of SSEBop and GG-NDVI estimates show satisfactory R-square and RMSE values. R-square values for SSEBop and GG-NDVI are 0.65 and 0.61, respectively. The results demonstrate that the ET estimates from GG-NDVI ET at an annual time-scale are reasonable. Figure 3, however, shows the 1:1 scatter of yearly variability of both models with GG-NDVI showing a tendency to underestimate in the higher ET range. In contrast, SSEBop tends to overestimate ET in the same higher ET range. Generally, higher ET occurs mostly in wet conditions, and underestimating




ET in moist regions is a characteristic of the complementary relationship (Han et al., 2014; Hobbins et al., 2001; Roderick et al., 2009).

Figure 4 shows the poor results of SSEBop with the temporal variation in $Th$, $Tc$, and $Ts$ on the left and the corresponding SSEBop, GG-NDVI, and measured ET values on the right. For example, at Austin Cary in Florida (Fig. 4(a)), RMSE ranged from 29 to 164 mm month⁻¹ for SSEBop and 17 to 70 mm month⁻¹ for GG-NDVI. Moreover, SSEBop showed significant deviations from measured ET throughout the year, and RMSE varied from 29 to 164 mm month⁻¹. Where SSEBop shows low RMSE values in Figs. 4(a) and 4(b), a possible reason for these significant deviations could be the concept of ET faction ($ETf$) in SSEBop. $ETf$ is calculated using $Th$, $Tc$, and $Ts$, and the $Ts$ curve lies mostly between the boundary conditions ($Th$ and $Tc$). However, $Ts$ in Fig. 4(a) is close to the predefined cold boundary ($Tc$), which brings $ETf$ closer to 1.0, resulting in a corresponding ET that is close to the maximum ET.

According to Table 1 and Fig. 5, the mean RMSE of GG-NDVI ranged between 15 and 20 mm month⁻¹, and GG-NDVI showed lower RMSE than SSEBop every year. Although the magnitude of agreement (overestimation or underestimation) seems to vary from site to site and from season to season, Figure 5 confirms that the occurrence of an RMSE less than 20 mm month⁻¹ with GG-NDVI is more frequent than with SSEBop in both dry and wet sites. The averages of RMSE across 24 dry sites for GG-NDVI and SSEBop are 19 mm month⁻¹ and 22 mm month⁻¹, respectively. For 36 wet sites, GG-NDVI and SSEBop showed an average RMSE of 17 mm month⁻¹ and 20 mm month⁻¹, respectively. These results indicate that GG-NDVI ET estimates improve with wetness, which is similar to the previous studies of Anayah and Kaluarachchi (2014), Hobbins et al. (2001), and Xu and Singh (2005).

Based on these results, we could conclude that GG-NDVI is a reliable approach for estimating ET, the novelty of GG-NDVI being that the Fu equation can be used to define relative evaporation in the original GG model using NDVI. This approach showed a reasonable match between GG-NDVI and the 60 AmeriFlux sites. However, GG-NDVI may not predict ET accurately when the vegetated cover changes significantly or is dense. For example, at Brooking in South Dakota, the mean RMSE of GG-NDVI was 42 mm month⁻¹, compared to 18 mm month⁻¹ with all sites, and NDVI has a large seasonal vegetation cover as shown in Fig. 6. A possible reason is that the relationship between NDVI and vegetation can be biased in sparsely vegetated areas with a Leaf Area Index (LAI) of less than 3. According to Pettorelli et al. (2005), the Soil Adjusted Vegetation Index (SAVI) is recommended instead of NDVI when LAI is less than 3. It should be noted that the LAI of Brookings is about 2.5. Furthermore, prior studies of Mu et al. (2011) and Yuan et al. (2010) have demonstrated that NDVI is insufficient to represent vegetation under dense vegetation conditions. Recently, Zhang et al. (2016) introduced the fraction of absorbed photosynthetic active radiation absorbed by vegetation (or fPAR) under the Budyko framework to avoid the bias of NDVI. Thus, this inability of NDVI to represent vegetation under dense conditions may be the reason for the decreased performance of GG-NDVI. Another possible reason according to Yang et al. (2009), the relative infiltration capacity and the average topographic slope need to be taken into consideration when using the Fu equation, especially in small catchments. Therefore, more work is needed to generalize the relationship for the use of NDVI with changing vegetation cover within the Budyko framework. The next section will discuss options to improve the GG-NDVI model.





### 3.2 Phase 2: Enhancements to GG-NDVI

As described earlier, GG-NDVI performed slightly better than SSEBop in both dry and wet climate conditions, and GG-NDVI increased the predictive power with increasing humidity. One interesting finding is that RMSE from GG-NDVI increases slightly with the relative evaporation parameter as shown in Fig. 7. Considering this observation, Phase 2 then

focused on the relationship between the performance of GG-NDVI and $G$ in the context of using the complementary relationship.

Within the complementary relationship, increasing $G$ means that climate is becoming wetter and ET is closer to ETW. When ET equals to ETW, surface has access to unlimited water as shown in Fig. 8. However, natural surfaces in even the wettest regions may not approach complete saturation, hence, ET can remain below its limiting value of ETW. Consequently, the

magnitude of difference between ET and ETW is important in estimating ET, especially under highly moist conditions. A possible explanation may be that the complementary relationship between ET and ETP with respect to ETW is not symmetric. GG-NDVI has improved the performance of the original GG model, but Eq. (3) still contains the value of 2, which refers to a symmetric complementary relationship. As explained earlier, other authors (Aminzadeh et al., 2016; Kahler and Brutsaert, 2006) question the use of a symmetric relationship. Thus, the use of a symmetric complementary relationship

may have contributed to the decreased performance of GG models, both the modified GG model and GG-NDVI. In order to understand the relationships affecting model accuracy, a correction function as a function of $G$ is required as shown in Eq. (17).

$$\text{ET} = \frac{2G_{new}}{G_{new}+1} \times f(G) \times \text{ETW} \tag{17}$$

where $f(G)$ is the correction function. We expect the correction function to be nonlinear, similar to an exponential function,

since the magnitude of the difference between ET and ETW decreases exponentially. In this work, we fitted 2772 data points to an exponential function similar to Eq. (18). Multiple regression analysis was conducted to compute the values of the $\alpha$ and $\beta$ coefficients.

$$f(G) = \alpha e^{\beta \cdot G} \tag{18}$$

Regression analysis found that $\alpha$ is 0.7895 and $\beta$ is 0.9655. Hereafter, the GG-NDVI model with the proposed correction

function given as Eq. (17) is called the Adjusted GG-NDVI model.

To determine the accuracy of Adjusted GG-NDVI, comparisons were made between the results from the Adjusted GG-NDVI and GG-NDVI and between measured ET data and ET values from SSEBop. These comparisons are shown in Fig. 9 and Table 2 across 60 sites. While ET from GG-NDVI at Mize in Florida (Fig. 9(a)) and Blodgett in California (Fig. 9(b)) showed deviations from measured ET, we can see that the Adjusted GG-NDVI produced ET estimates close to measured ET

and reduced mean RMSE from 33 to 22 mm month[-1] for Mize and 17 to 10 mm month[-1] for Blodgett. In Table 2, overall RMSE across 60 sites for GG-NDVI and Adjusted GG-NDVI were found to be 18 mm month[-1] and 15 mm month[-1], respectively. Figure 10, which presents a histogram of RMSE from the different ET models, shows a significant improvement attributed to the Adjusted GG-NDVI model. With Adjusted GG-NDVI, 38 sites have less than 15 mm month[-1]




of RMSE, compared to 26 sites with GG-NDVI. These results suggest that the use of the correction function in GG-NDVI can significantly improve accuracy in estimating ET. In addition, Eq. (17) can be updated with the new definition of $G$ as

$$\text{ET} + \text{ETP} = 2f(G)\text{ETW} \tag{19}$$

where the value of $2f(G)$ can vary between 1.64 and 3.04 as $G$ varies based on site-specific conditions. The new formulation of the Adjusted GG-NDVI model described in Eq. (19) clearly shows that the relationship between ET and ETP is not symmetric with respect to ETW, further confirming the earlier conclusions that the hypothesis of Bouchet (1963) needs to be extended and applied with appropriate corrections.

## 4 Summary and Conclusions

ET estimation models using the complementary relationship are able to estimate ET in most instances. In particular, the model proposed by Anayah and Kaluarachchi (2014) showed excellent performance compared to recently published studies. However, the predictive power of this model and other similar models decreases with increasing aridity (Anayah and Kaluarachchi 2014; Hobbins et al., 2001; Xu and Singh, 2005). In the case of the modified GG model proposed by Anayah and Kaluarachchi (2014), a reason may be that relative evaporation in the original GG model was derived using 158 sites in Canada under mostly humid conditions. To overcome this limitation, the previously revised GG model, GG-NDVI (Kim and Kaluarachchi, 2017), used the Fu equation to describe relative evaporation on the basis that the Budyko framework can support the complementary relationship (Zhang et al., 2004; Yang et al., 2006). The results of GG-NDVI showed improved accuracy compared to other complementary relationship models but also showed the need for further refinements, especially under dense vegetation conditions. On the other hand, remote sensing methods are more common as operational models under field conditions. In order to determine whether complementary methods such as GG-NDVI can compete and deliver accuracy similar to remote sensing methods, it is important make appropriate comparisons. The objectives of this work were therefore twofold: (1) evaluate the recently developed ET estimation method, GG-NDVI, to see if it could deliver similar accuracy to the commonly used operational remote sensing method, SSEBop and (2) identify the inherent weaknesses of the original complementary relationship and make appropriate refinements to further improve the GG-NDVI model, especially under dense vegetation conditions. For this purpose, we selected 60 AmeriFlux sites located across the US.

The first phase of the analysis showed that the GG-NDVI model with the Budyko framework and relative evaporation was found to work reasonably well. Validation with 60 AmeriFlux sites indicated similar levels of accuracy for both SSEBop and GG-NDVI. R-square between GG-NDVI and measured ET ranged from 0.40 to 0.79, overall RMSE of GG-NDVI ranged between 15 and 20 mm month[-1], and GG-NDVI showed lower RMSE than SSEBop every year. Furthermore, the occurrences of RMSE less than 20 mm month[-1] with GG-NDVI were more frequent than SSEBop. Based on these results, we concluded that GG-NDVI is a reliable approach for estimating ET.

The second phase of the analysis showed that the predictive power of GG-NDVI decreased with relative evaporation possibly due to the use of the symmetric complementary relationship in estimating ET. In order to identify the true



relationship between ET and ETP with respect to ETW, an exponential correction function was proposed. This phase demonstrated that the inclusion of relative evaporation with a correction function greatly improved the performance of the Adjusted GG-NDVI. For example, 68 % of Adjusted GG-NDVI sites had RMSE less than 15 mm month$^{-1}$ compared 43 % with GG-NDVI.

In essence, this study strengthens the idea that the use of vegetation cover information in the complementary relationship has increased ET estimation power. More importantly, this work showed that the symmetric relationship typically assumed with the complementary relationship may not be valid.  Instead, the results show that the symmetrical relationship needs to be updated with a nonlinear correction function as proposed here. A key strength of this study is that the latest proposed version of the GG model, Adjusted GG-NDVI, overcomes limitations of both relative evaporation as proposed by Granger and Gray

(1989) and the assumption of a symmetric complementary relationship from the work of Bouchet (1963). Consequently, Adjusted GG-NDVI can lead to significantly increased accuracy of ET estimates under diverse climate conditions while producing comparable or even better results than the SSEBop operational remote sensing model.

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





**Table 1. Comparison of monthly ET estimates between SSEBop and GG-NDVI using AmeriFlux data from 2000 to 2007.**

| Year | AmeriFlux mean [mm month$^{-1}$] | R-square | | RMSE [mm month$^{-1}$] | |
|------|------|------|------|------|------|
| | | SSEBop | GG-NDVI | SSEBop | GG-NDVI |
| 2000 | 43 | 0.82 | 0.79 | 16 | 15 |
| 2001 | 44 | 0.54 | 0.58 | 23 | 20 |
| 2002 | 41 | 0.73 | 0.67 | 19 | 16 |
| 2003 | 42 | 0.68 | 0.65 | 21 | 17 |
| 2004 | 42 | 0.68 | 0.60 | 18 | 18 |
| 2005 | 42 | 0.37 | 0.57 | 28 | 18 |
| 2006 | 41 | 0.61 | 0.55 | 20 | 18 |
| 2007 | 34 | 0.40 | 0.40 | 18 | 17 |
| All years | 44 | 0.65 | 0.61 | 19 | 18 |





**Table 2. Comparison of RMSE between GG-NDVI, SSEBop, and Adjusted GG-NDVI across 60 sites.**

| ET Model | RMSE [mm month$^{-1}$] | | |
| --- | --- | --- | --- |
| | Min | Mean | Max |
| GG-NDVI | 7 | 18 | 48 |
| SSEBop | 8 | 20 | 48 |
| Adjusted GG-NDVI | 7 | 15 | 34 |



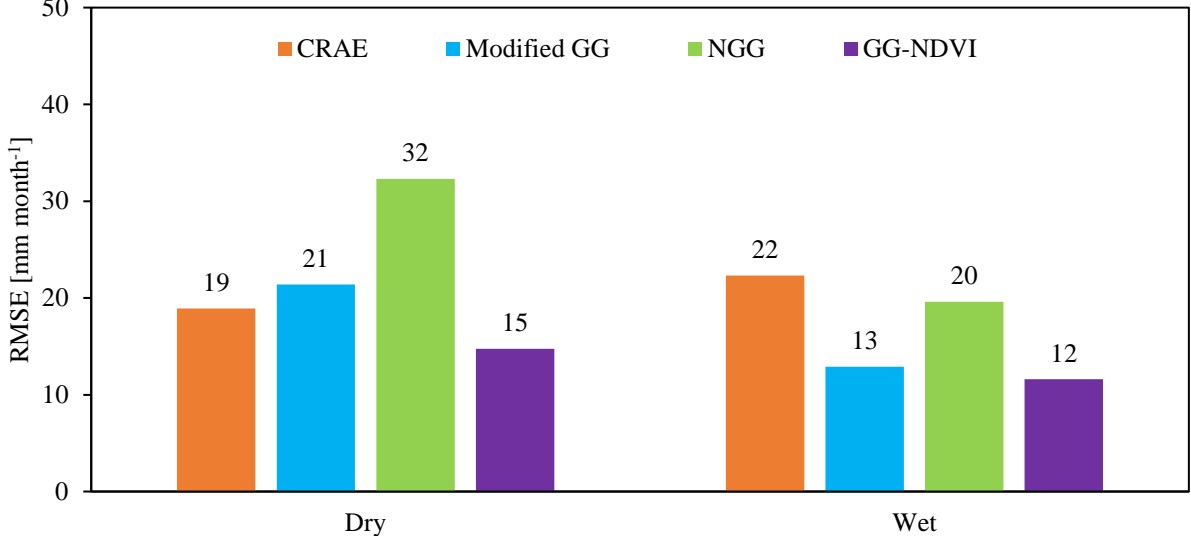

**Figure 1. Comparison of RMSE (Root Mean Squar Error, mm month$^{-1}$) between different complementary relationship models for 29 dry and 30 wet sites in the US. NGG and GG-NDVI refer to the models of Han et al. (2011) and Kim and Kaluarachchi (2017), respectively.**



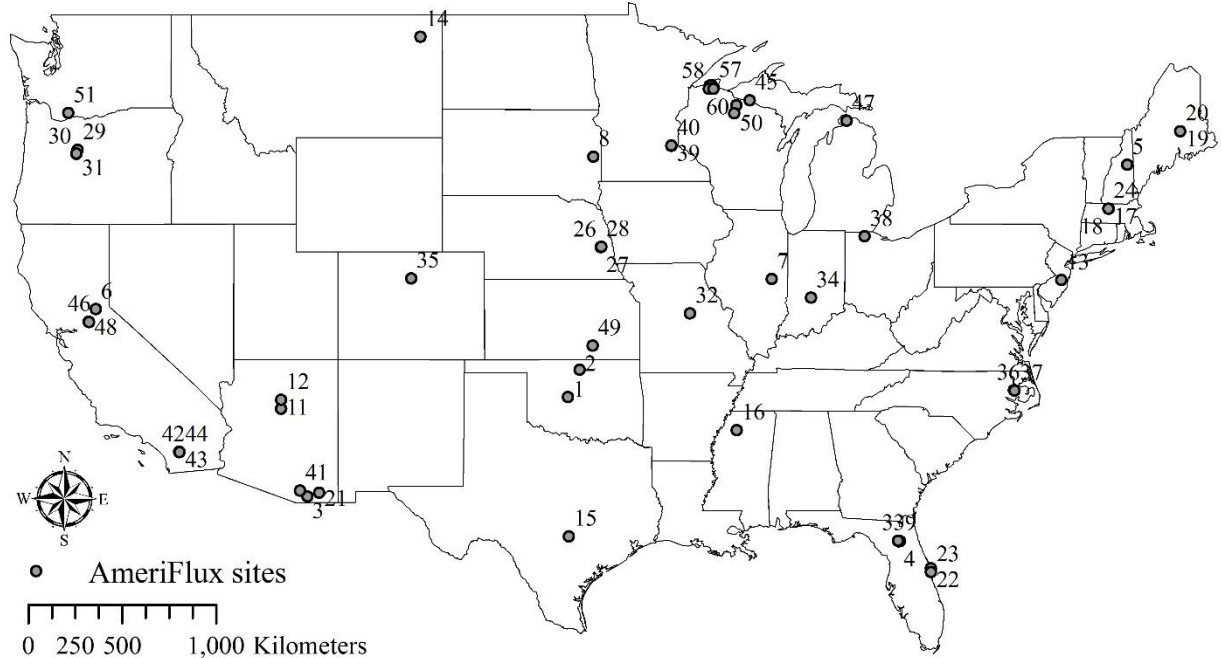

Figure 2. Locations of 60 AmeriFlux Eddy Covariance sites used in this study with number.





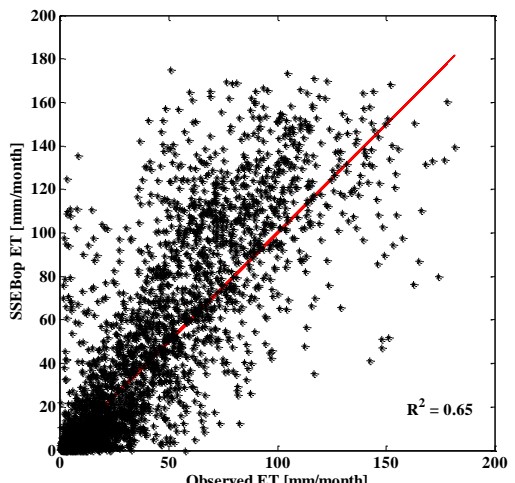
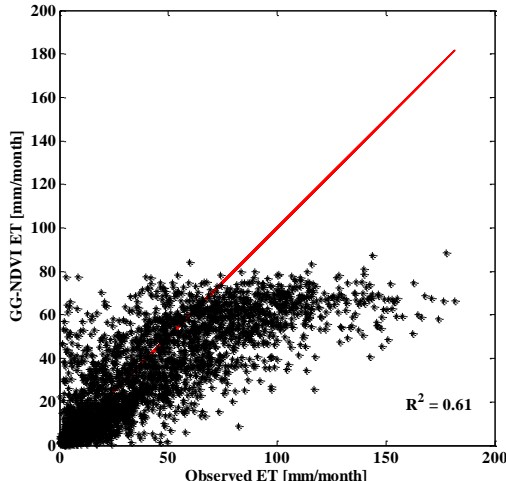

**Figure 3. Validation results of monthly ET estimates from SSEBop and GG-NDVI against AmeriFlux ET data between 2000 and 2007.**



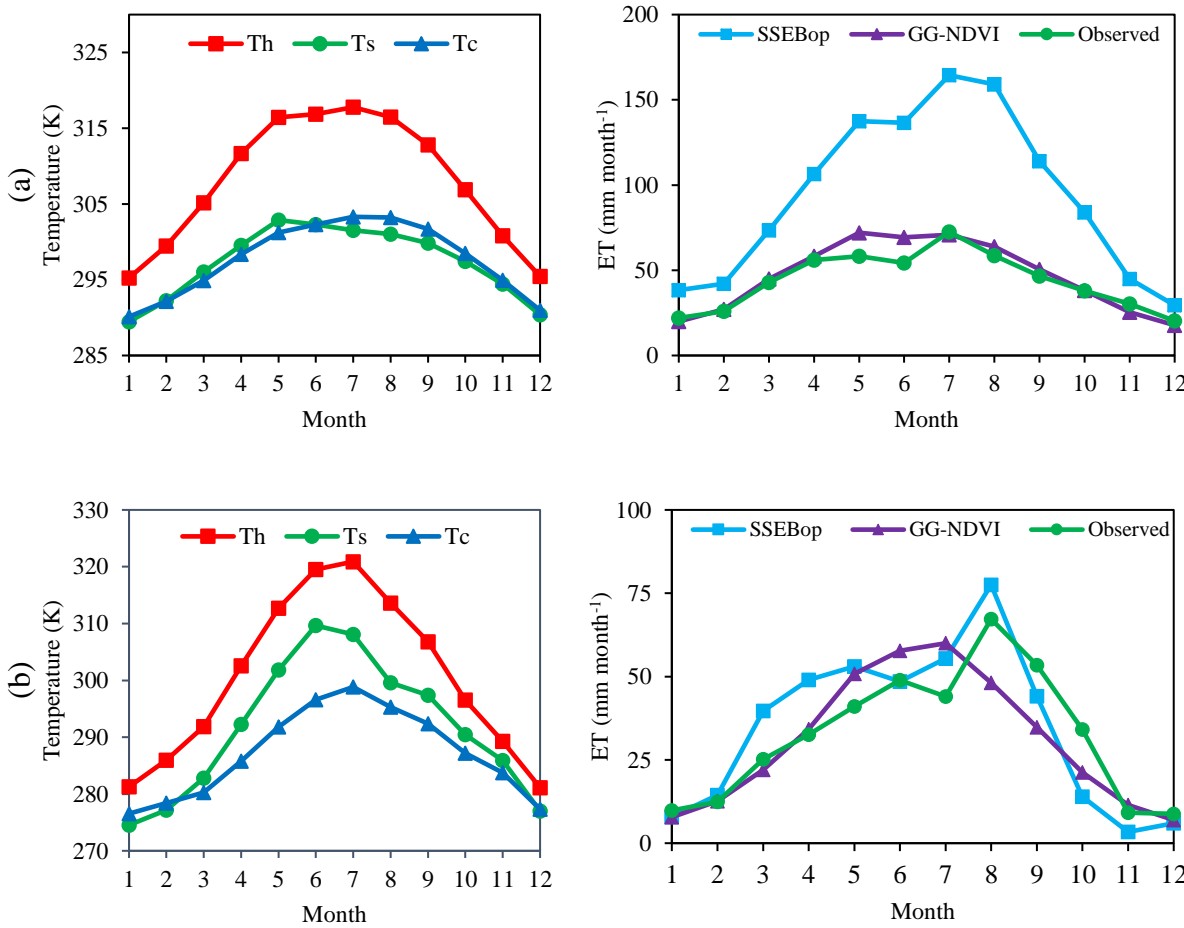

**Figure 4. Temporal variation of 8-day average Ts, Th, Tc (left) and monthly ET estimates from SSEBop and GG-NDVI and measured ET at (a) Austin Cary in Florida and (b) Flagstaff in Arizona for 2005.**



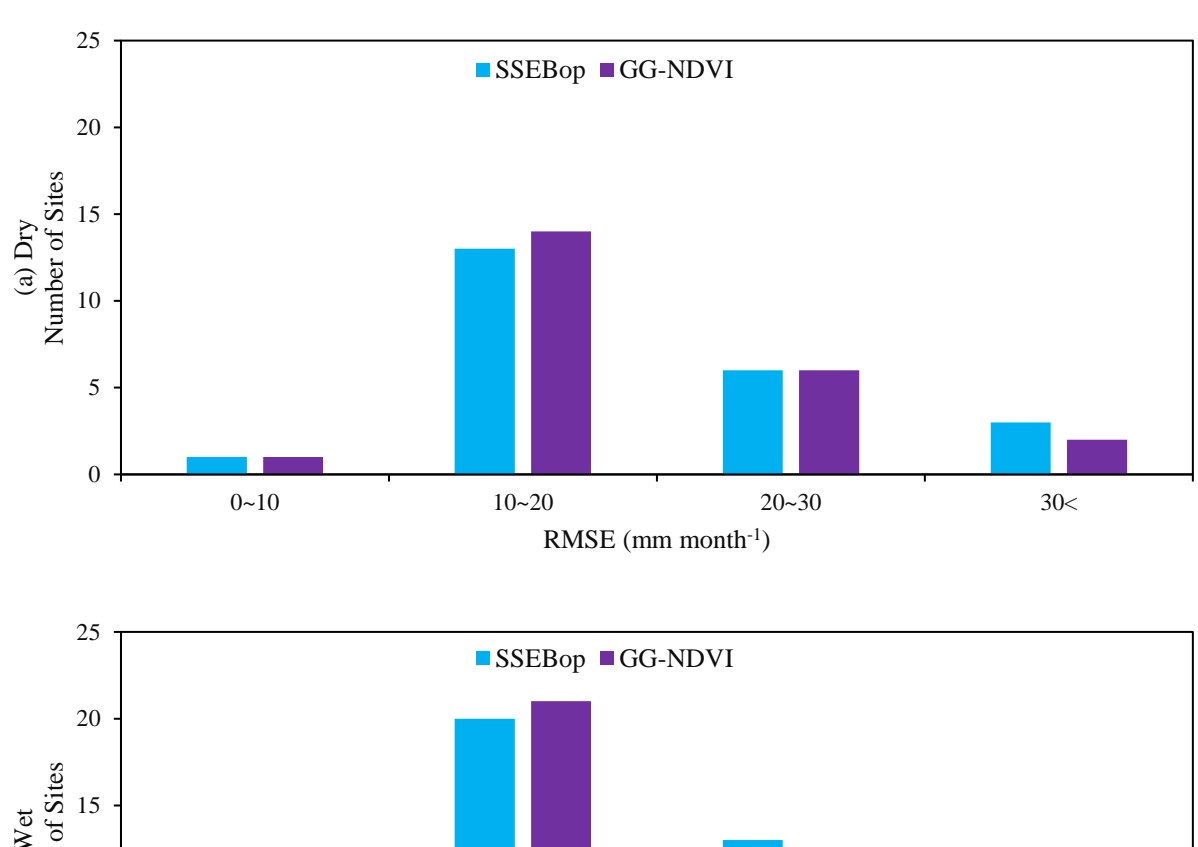

**Figure 5. Histogram of RMSE values of SSEBop and GG-NDVI for (a) dry and (b) wet sites.**



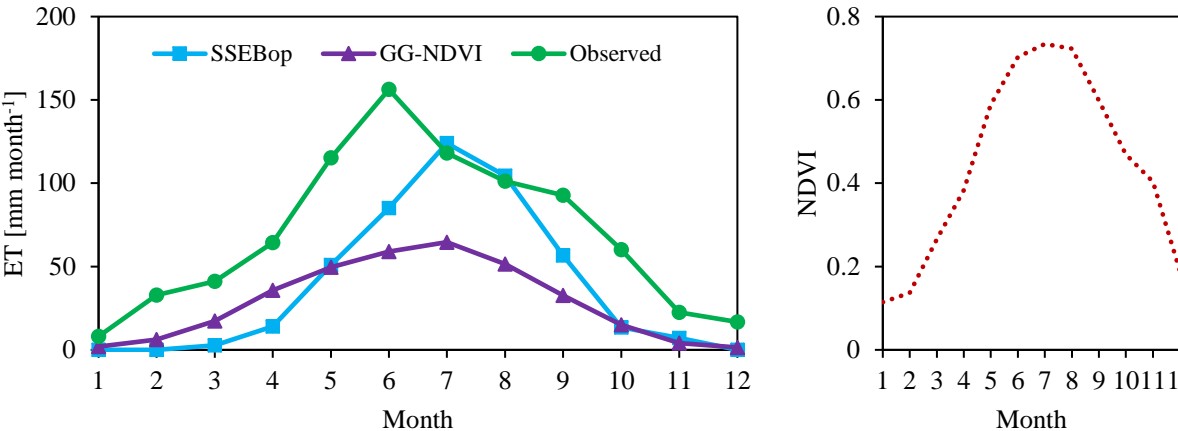

**Figure 6. Comparisons of monthly ET between SSEBop and GG-NDVI against measured ET (left) and time-series of NDVI at Brookings in South Dakota (right).**





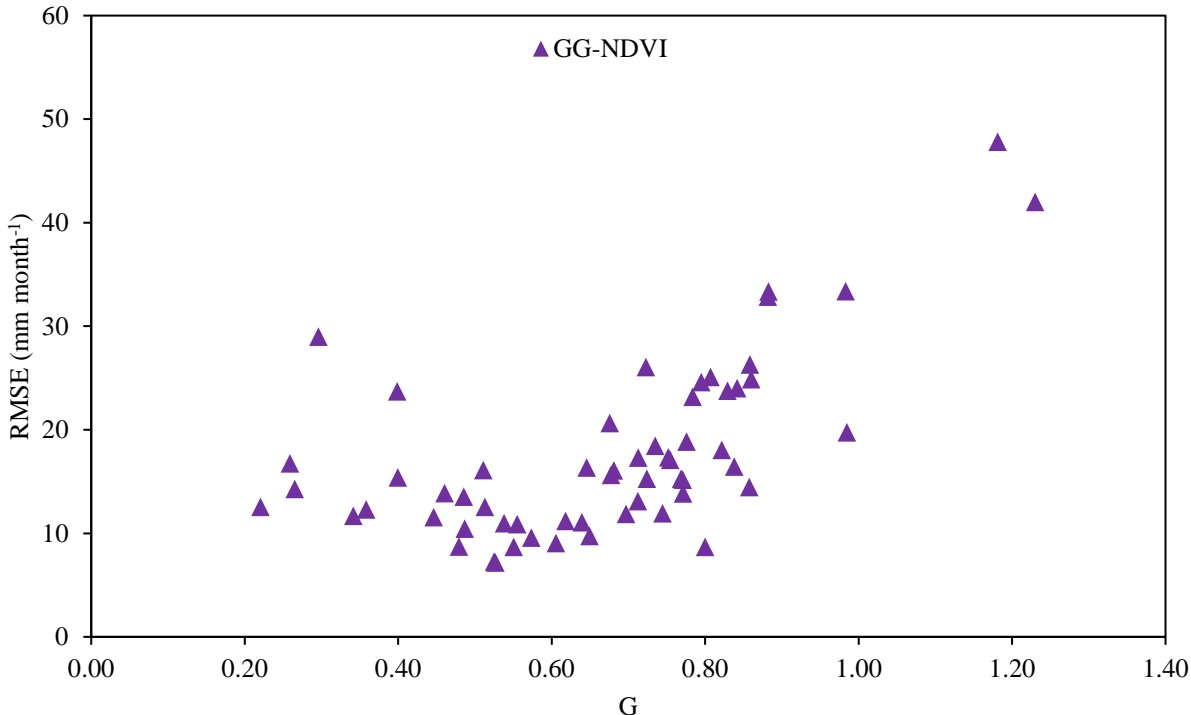

**Figure 7. RMSE from GG-NDVI versus relative evaporation ($G$ = ET/ETP).**





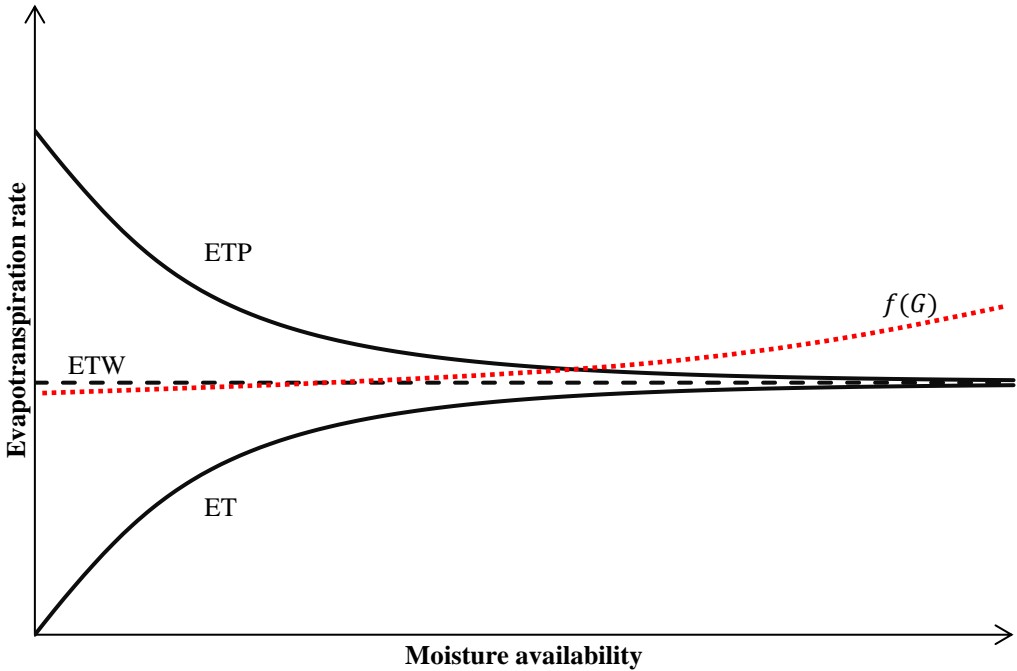

**Figure 8. A schematic representation of the complementary relationship between ET, ETP, and ETW with the proposed correction function, $f(G)$.**





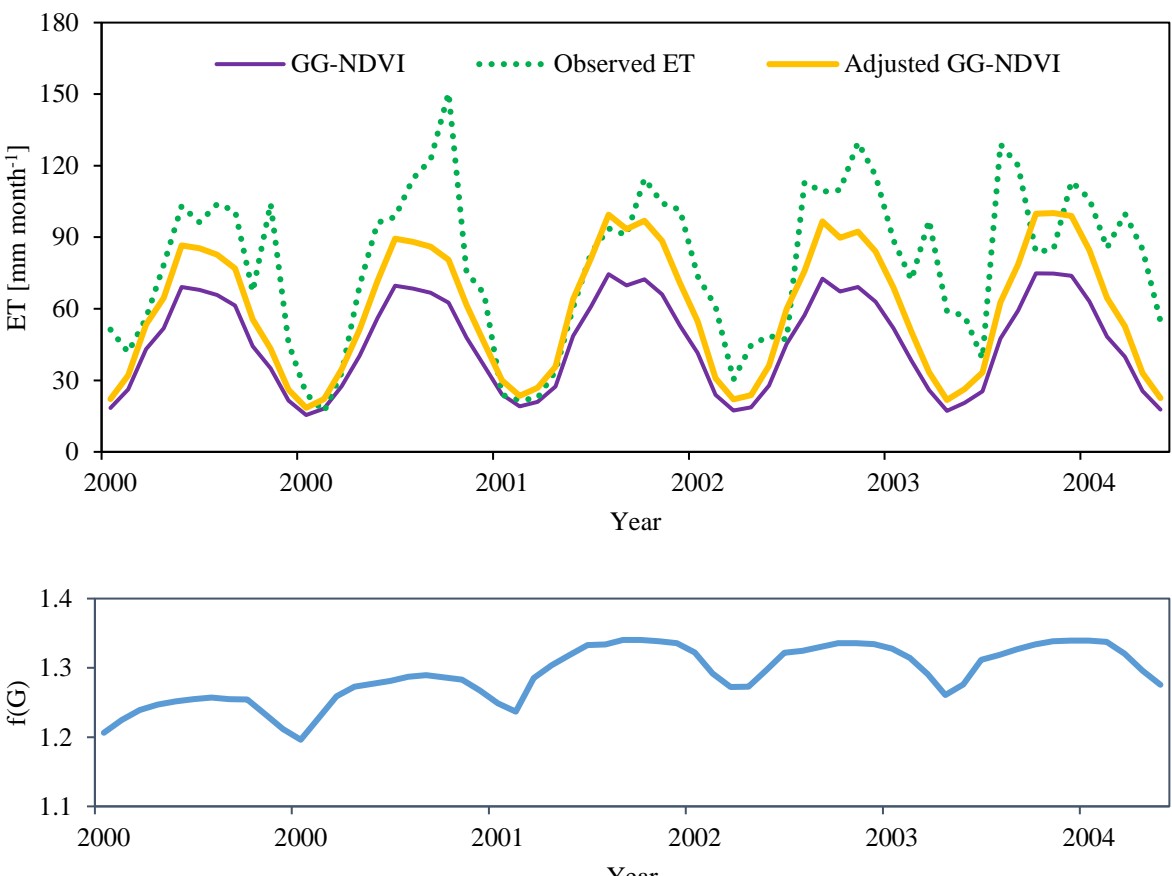

**Figure 9(a). Comparison of monthly ET values of GG-NDVI and Adjusted GG-NDVI with measured ET and the corresponding**
5 $f(G)$ **at Mize, Florida from 2000 to 2004.**





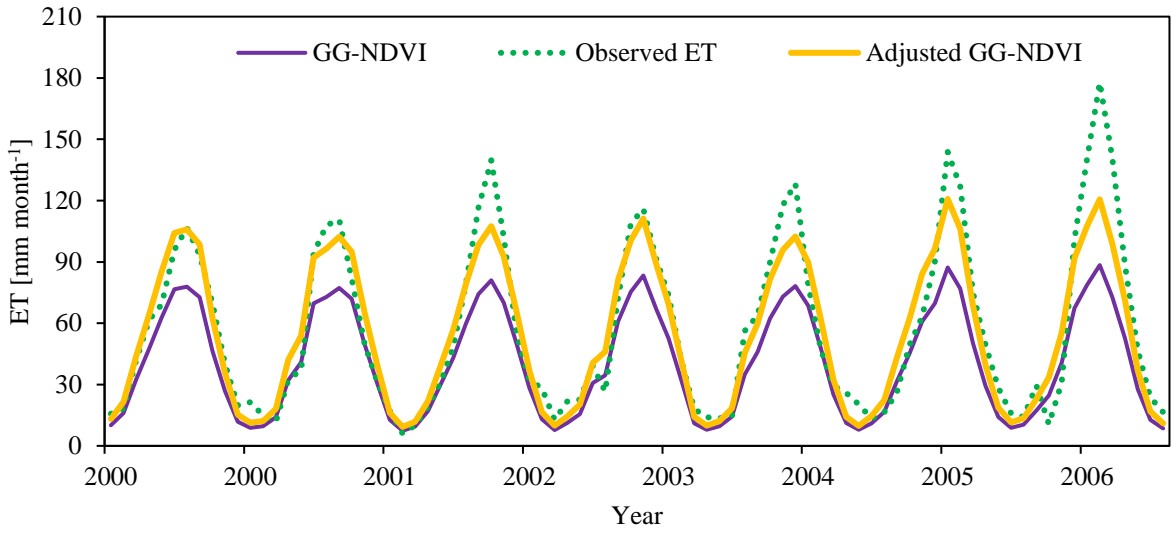

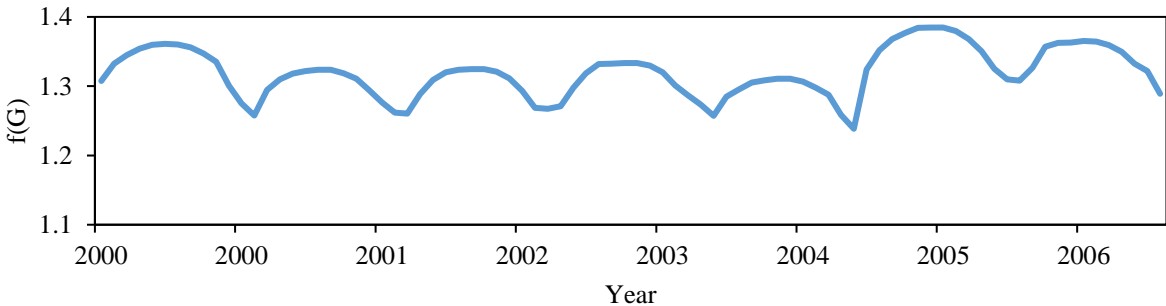

**Figure 9(b). Comparison of monthly ET values of GG-NDVI and Adjusted GG-NDVI with measured ET and the corresponding** $f(G)$**at Blodgett, California from 2000 to 2006.**





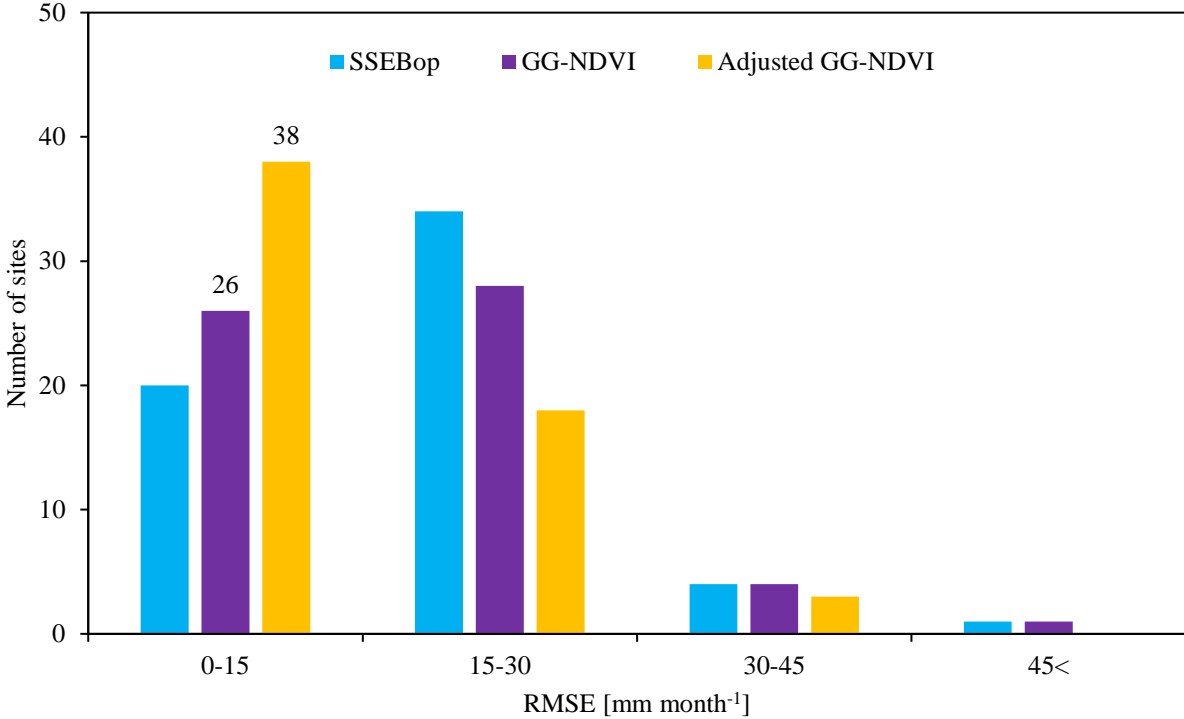

**Figure 10. Comparison of RMSE values between different ET estimation models.**