# Peer review of "Complementary Relationship for Estimating Evapotranspiration Using the Granger-Gray Model: Improvements and Comparison with a Remote Sensing Method"

_Hydrology and Earth System Sciences, 2017_

## Short Comment (SC1) · 25 Aug 2017

Review of "Complementary relationship for estimating evapotranspiration using the Granger-Gray model" by H. Kim and J. J. Kaluarachchi

The authors use an outdated formulation (Granger and Gray, 1989) of the Complementary Relationship (CR) of evaporation (Bouchet, 1963) with complete disregard of the recent developments in the area (Brutsaert, 2015; Crago et al., 2016; Szilagyi et al., 2017). They use vegetation indexes taken in sunny days to derive parameters of their

model to predict monthly ET rates and validate it with another model (SEBBop) which is absolutely irrelevant plus with Ameriflux data. The question of symmetry or not of the CR is outdated too in the light of the above recent developments of CR research. Beside the three original model parameters (alpha, a, b) they introduce additional ones, k, beta to improve the prediction capacity of their model, when there exists a calibration-free version of the CR (i.e., Szilagyi et al., 2017), perhaps performing even better. The most problematic however is the validation of their modelled fluxes with Ameriflux data. For example I happen to know the Nebraska Ameriflux sites, which are made up of different crop fields. Probably a great number of other Ameriflux sites are similar in horizontal extent. The CR is valid for regional evaporation studies not for ET rates over plot-sized areas in a heterogeneous setting, so this way the authors are comparing apples and oranges. No wonder they can never get the largest measured ET rates with their model in spite of the artificial tuning of the parameters. This scale issue is so important that I cannot recommend the work for publication due to its mismatched validation effort and the complete disregard of recent very important developments in CR studies.

New references: Brutsaert, W. 2015 A generalized complementary principle with physical constraints for land-surface evaporation. Water Resources Research, 51, doi:10.1002/2015WR017720. Crago, R., Szilagyi, J., Qualls, R. J. & Huntington, J. 2016 Rescaling the complementary relationship for land surface evaporation. Water Resources Research, 52, doi:10.1002/2016WR019753. Szilagyi, J., Crago, R. & Qualls, R. J. 2017 A calibration-free formulation of the complementary relationship of evaporation for continental-scale hydrology. Journal of Geophysical Research Atmospheres, 122, doi: 10.1002/2016JD025611.

---

## Referee Comment (RC2) · Anonymous Referee #2 · 29 Aug 2017

In this paper authors present an assessment of the accuracy of the Granger-Gray model to compute actual evapotranspiration and propose a new calibration of coeficients. It is not clear why authors include SSEBop evapotranspiration estimates. In my opinion these are only values estimated with a different model. So I think real validation should be done by comparing the Granger-Gray model estimates with evapotranspiration fluxes measured at eddy-covariance sites. Regarding these latter, authors should check that evaporation fluxes are representative of the vegetation type they are considering. So they should check the station footprint in order to detect possible fluxes from

heterogeneous vegetation that should be discarded from analysis. The new formulation of GG equation with a f correction function has limited usefulness as it is only valid for the territory and the period analysed in this study. Of course it is a large territory but, in order to demonstrate validity and robustness of this equation, I suggest to split the dataset into two parts. On the first part authors should calibrate parameters of the correction function and they should use the second part to validate the new equation.

———————————————

---

## Author Comment (AC1) · 29 Aug 2017

Homin Kim and Jagath J. Kaluarachchi

homin.kim@aggiemail.usu.edu

ET estimation methods can be divided into two categories: (1) ground-based ET that use standard meteorological data; and (2) ET models that based on remote sensing data that must be combined with retrieval algorithms to estimate ET in accordance with McMahon et al. (2016). From the study of Kim and Kaluarachchi (2017), the model without a 'f' function was validated by comparing with other complementary relationship models including CRAE, GG, AA, and Modified GG (Anayah and Kaluarachchi, 2014) with the 59 eddy-covariance sites measured ET. The mean RMSE of our model

across the 59 sites was 14 mm month-1 compared to 21 mm month-1 of CRAE, 28 mm month-1 of AA, 27 mm month-1 of GG, and 17 mm month-1 of the Modified GG. We showed these comparison results presented in Figure 1 without GG and AA models because the validation of our model with original ET methods was the purpose of the previous study of Kim and Kaluarachchi (2017). For further information, the Modified GG model also was validated with CRAE, AA, and GG across 34 global sites. Since these findings are good within the first category (ground-based ET), the subject of this study was to validate our model with the widely used remote sensing method (SSE-Bop, Senay et al. 2013) that was developed by USGS. We agree with the evaporation fluxes and validation comments. For the evaporation comment, we collected the level 4 measured meteorological data and latent heat flux (LE) data at 76 AmeriFlux towers then we excluded those towers with actual vegetation type different from MODIS land cover type at any surrounding 3x3 km2 pixels. Then, we further excluded those towers with less than half a year of measurements during 2000-2007. Finally, 60 sites were involved in this study. We also know about the limitation of the NDVI. Several studies (Pettorelli et al. 2005, Yuan et al. 2010, Mu et al. 2011) recommended instead of NDVI for heterogeneous areas with LAI (Leaf Area Index) and SAVI (Soil-Adjusted Vegetation Index). However, these indices also have limitations that they require more data. For the most efficient process, we adopted the NDVI with theoretical backgrounds (Yang et al. 2006, Zhang et al. 2004). More research is needed to better understand the heterogeneous vegetation. Also, further validation is also needed to establish the robust equation as the comment. In this study, we attempted to demonstrate the limitation of complementary relationship which refers to a symmetric relationship and to introduce the enhance method with a correction function. Although the current study is based on 60 eddy covariance sites, the further study is regarding the other data sources including remote sensing data across the United States to follow the suggestion and could assess the potential applications of ET.

346, 2017.

---

## Author Comment (AC3) · 4 Sep 2017

Homin Kim and Jagath J. Kaluarachchi

homin.kim@aggiemail.usu.edu

The method we used in this study was not outdated. The starting point of this study was further refinement of the study of Anayah and Kaluarachchi (2014). Their ET method applied to estimate groundwater recharge in Ghana (Anayah et al., 2013) and assess the drought condition across the United States (Kim and Rhee, 2016). Meanwhile, we developed the GG-NDVI model to overcome the limitation of Anayah and Kaluarachchi (2014) and validation with other CR methods including CRAR, GG, and AA models was done in Kim and Kaluarachchi (2017). Additionally, the

Granger and Gray (1989) model is still ongoing research. For example, Zhu et al. (2016) evaluated the GG model using 12 eddy covariance flux stations in China. The purpose of this study was further validation with the widely used remote sensing method, SSEBop, developed by USGS (Senay et al., 2013). According to Senay et al. (2013), SSEBop was validated using 45 eddy covariance stations in the United States and Velpuri et al (2013) also used 60 eddy covariance stations in AmeriFlux for a comprehensive evaluation of MODIS global ET and SSEBop. Out validations were in line with these studies. We collected the level 4 measured meteorological data and latent heat flux (LE) data at 76 eddy covariance AmeriFlux stations then, we excluded those stations with actual vegetation type different from MODIS land cover type at any surrounding 500 m by 500 m pixels. Then, we further excluded those stations with less than half a year of measurements during 2000-2007. Finally, 60 sites were involved in this study without possible fluxes from heterogeneous vegetation. The manuscript is modified in Lines 3-7 at Page 8. Importantly, we did not disregard the recent study of Szilagyi et al. (2017) and Crago et al. (2016). Then, Crago et al. (2016) model was calibrated their parameters. They mentioned that the coefficients, a0, a1, a2, and a3, may be selected on physical grounds or through calibration and these values have been calibrated for their study and no additional requirement beyond those for any application of CR methods. Also, Szilagyi et al. (2017) noted that calibration of the parameters alpha and s0 were performed by a systematic trial and error approach and the objective function of calibration consisted of minimizing the RMSE between mean annual ET estimates and water balance derived ones. This process was similar to our model's. The Omega value in Eq. (7) was derived through a curve fitting procedure that minimizes RMSE between the measured and predicted evaporation ratio as mentioned in Line 16-17 at Page 6. Moreover, our model's parameters, alpha and beta in Eq. (18), do not need calibration anymore beyond this study such as Crago et al. (2017) mentioned. Also, k in Eq. (14) is constant and a recommended value of k is 1.2 for the United States. We excluded the additional sentence of Line 23-24 at Page 7 to avoid confusion and added studies of Szilagy et al. (2017) and Crago et al.

(2016) as references. Please see Lines 15 and 31-32 at Page 2. Unfortunately, we did not have much time to compare and review the study of Szilagy et al. (2017) because the current study was done before Szilagy et al. (2017) published. As the reviewer mentioned, it would be interesting to assess the scale issue and compare with the calibration-free version of CR. This is future research worthy of consideration.

Please also note the supplement to this comment:
https://www.hydrol-earth-syst-sci-discuss.net/hess-2017-346/hess-2017-346-AC3-supplement.pdf

**Supplement:**

[revised manuscript text omitted]

---

## Author Comment (AC4) · 6 Sep 2017

The method used in this study is not outdated and we disagree with this comment. The starting point of this study was further refinement to the model proposed by Anayah and Kaluarachchi (2014). Their ET model was successfully applied to estimate groundwater recharge in Ghana (Anayah et al., 2013) and in a follow-up study to assess the drought conditions across the United States by Kim and Rhee (2016). Therefore the comment to state that this model is outdated is incorrect. Meanwhile, we further improved the GG-NDVI of Anayah and Kaluarachchi (2014) to better accommodate arid

conditions and provided further validation compared with other CR methods; such as CRAY, GG, and AA models. These efforts were clearly demonstrated in an earlier publication by Kim and Kaluarachchi (2017). Additionally, the Granger and Gray (1989) model is still an important model in today's research (Zhu et al. 2016, Gao et al. 2016) and no means an older model.

The purpose of this study is to further demonstrate the validity of the earlier ET model of Kim and Kaluarachchi (2017) using the widely used USGS remote sensing model, SSEBop (Senay et al., 2013). According to Senay et al. (2013), SSEBop was validated using 45 eddy covariance stations in the United States while Velpuri et al (2013) used 60 eddy covariance stations of AmeriFlux for a comprehensive evaluation of MODIS global ET and SSEBop. Therefore this work follows clear and consistent validation in line with the earlier studies. We collected level 4 measured meteorological data and latent heat flux (LE) data from 76 eddy covariance AmeriFlux stations. Thereafter, we excluded those stations with actual vegetation type different from MODIS land cover type from locations of 500 m by 500 m pixels. We also excluded those stations with less than half a year of measurements during 2000-2007. Finally, 60 sites were selected and used in this study without heterogeneous vegetation conditions.

It should be noted that we did not disregard the recent study of Szilagyi et al. (2017) and Crago et al. (2016). In the latter study, they mentioned that coefficients, a0, a1, a2, and a3 in their model may be selected based on physical basis or through calibration. In their work, these values were obtained through model calibration. Szilagyi et al. (2017) noted that calibration of parameters alpha and s0 were performed by a systematic trial and error approach and the objective function of calibration consisted of minimizing RMSE between mean annual ET estimates and ET values derived from water balance. This process is similar to the work proposed by us. The Omega value in Eq. (7) was derived through a curve fitting procedure that minimizes RMSE between the measured and predicted evaporation ratios as mentioned in Line 16-17 at Page 6. Moreover, our model's parameters, alpha and beta in Eq. (18), do not need further calibration in the

application to other study areas and or future applications.

k in Eq. (14) is a constant and the recommended value is 1.2 for United States. We excluded the additional sentence of Line 23-24 at Page 7 to avoid confusion and added studies of Szilagy et al. (2017) and Crago et al. (2016) as references. Please see Lines 15 and 31-32 in Page 2. Unfortunately, we did not have much time to compare and review the study of Szilagy et al. (2017) because the current study was done before Szilagy et al. (2017). As the reviewer mentioned, it would be interesting to assess the scale issue and compare with Szilagy et al. (2017) for a possible future study.

New References for reviewer

1) Anayah, F. M., Kaluarachchi, J. J., Pavelic P., and Smakhtin, V.: Predicting groundwater recharge in Ghana by estimating evapotranspiration, Water International, 38, 4, 408-435, http://dx.doi.org/10.1080/02508060.2013.821642, 2013. 2) Kim, D., and Rhee, J.: A drought index based on actual evapotranspiration from the Bouchet hypothesis, Geophys. Res. Lett., 43, 10, 277-10,285, doi:10.1002/2016GL070302, 2016. 3) Gao, Y., Gan, G., Liu, M., Wang, J.: Evaluating soil evaporation parameterizations at near-instantaneous scales using surface dryness indices, Journal of Hydrology, 541, 1199-1211, 2016. 4) Zhu, G.-F., et al., Evaluating the complementary relationship for estimating evapotranspiration using the multi-site data across north China. Agric. Forest Meteorol., http://dx.doi.org/10.1016/j.agrformet.2016.06.006, 2016.

---

## Referee Comment (RC3) · Anonymous Referee #3 · 18 Sep 2017

I am not sure what to make of this paper. The authors aim to show that a previously developed method based on a modified Granger-Gray model (by the same authors published in 2017 in the Journal of Water and Climate Change) fits well with flux data and can be further improved by taking account of the a-symmetry of the complementary relationship at higher land surface wetness. But there are a number of fundamental issues that remain.

1) Why did they include the comparison with the remote sensing method? It serves no purpose. It certainly does not reveal what is wrong with the GG-NDVI method for

wetter circumstances? This can solely be derived from comparison with the flux data.

2) It seems that the only incremental advance in the paper is the f(G) function that corrects for wetter circumstances. This seems to warrant a technical note only, whereby many parts of the paper (derivation of the GG-NDVI method, all the remote sensing stuff, as well as the review of methods in the introduction) are unnecessary.

3) The f(G) function itself is not based on sound physical reasoning. In Phase 1, changing NDVI values over time are said to cause the larger errors, while in Phase 2 it is said that even at saturation E will remain smaller than EW and EW is increased. i.e. by multiplying with a function f(G) that is empirically determined and thus necessary includes many effects.

One small issue: - The authors state that the GG model is not suitable for drier circumstance because it was only tested for wetter circumstances in Canada. But that is not what is said in the Granger-Gray 1989 paper. In fact, the opposite! First, they refer to the data coming from the semi-arid climate zone of Western Canada. Second, if one looks at Figure 2 in that paper we see that the relationship is fitted to G values smaller than 0.6, which means it is most suitable for dry circumstances.

---

## Author Comment (AC5) · 29 Sep 2017

Reviewer comment 1. Why did they include the comparison with the remote sensing method? It serves no purpose. It certainly does not reveal what is wrong with the GG-NDVI method for wetter circumstances? This can solely be derived from comparison with the flux data. → There are two types of ET models and these are ground-based and remote sensing (RS)-based per McMahon et al. (2016). Among the ground-based ET models, GG-NDVI can be considered as one of the most effective ET models (Kim & Kaluarachchi, 2017). For RS-based ET, SSEBop is widely used and have been validated by many studies including USGS (Velpuri et al., 2013; Senay et al., 2013). Since 2016, USGS Geo Data Portal provides SSEBop ET data across the United States at 1-km resolution from 2000 to 2015. With advanced remote sensing techniques, RS-based ET models are gradually accepted as operational and effective compared to ground-based models. Therefore, the first objective of this study is the demonstrate the validity of the GG-NDVI model in comparison with this commonly used operational model, SSEBop. Accordingly, the GG-NDVI model showed similar accuracy or even better results than the SSEBop model. The major reason to move from GG-NDVI to Adjusted GG-NDVI is because certain results still showed questionable shift (see Figure 7). A careful investigation showed the cause of this departure is the assumption of linear complementary relationship. When we removed this linear assumption with an appropriate nonlinear function f(G), the results improved dramatically confirming the fact that the linear assumption is not always valid. As the reviewer mentioned, the limitation of the symmetric relationship was derived from comparison with flux data, and the subsequent model, Adjusted GG-NDVI, was validated using the remote sensing method. In brief, both GG-NDVI and Adjusted GG-NDVI were validated using SSE-Bop and the results indicated that the Adjusted GG-NDVI model can provide accurate ET estimations under diverse climate conditions while producing better results than SSEBop.

Reviewer comment 2. It seems that the only incremental advance in the paper is the f(G) function that corrects for wetter circumstances. This seems to warrant a technical note only, whereby many parts of the paper (derivation of the GG-NDVI method, all the remote sensing stuff, as well as the review of methods in the introduction) are unnecessary. → We found the limitation of GG-NDVI from Figure 7 and this limitation is related to the symmetric relationship between ET and ETP or (ETW). The question of symmetry of the complementary relationship was raised by prior researchers as well (Kahler & Brutsaert, 2006; Venturini et al., 2011). Therefore, we were not ready to move ahead with the GG-NDVI as the final model as the results showed questionable behavior under selected conditions. Therefore, as mentioned earlier, Adjusted GG-

NDVI is the resulting and improved model. Note we found the weak performance of GG-NDVI with increasing G from the Figure 7. The corresponding correction function, f(G), was developed using 2,772 data points from 60 flux sites covering wide range of conditions. As a result, the Adjusted GG-NDVI model performed well for all 60 sites. With the introduction of the correction function f(G), our understanding of the complementary relationship increased significantly while providing much improved and accurate ET estimates.

Reviewer comment 3. The f(G) function itself is not based on sound physical reasoning. In Phase 1, changing NDVI values over time are said to cause the larger errors, while in Phase 2 it is said that even at saturation E will remain smaller than EW and EW is increased. i.e. by multiplying with a function f(G) that is empirically determined and thus necessary includes many effects. → Within the complementary relationship, ETW is not increased by using f(G). Theoretically, the symmetric relationship will be changed. Exclusion of f(G) in Eq. (19) brings it back to the original form of the complementary relationship. The value of '2' denotes the symmetric relationship between ET and ETP while f(G) removes this assumption and make it nonlinear. Using f(G) in Eq. (19) means that there is no exact shape for the relationship between ET and ETP. As we mentioned in the manuscript, the nonlinear exponential function was used because the difference between ET and ETW (or ETP) decreases exponentially with increasing wetness and the approach used here is similar to the study of Kahler & Brutsaert (2006).

Minor comment. The authors state that the GG model is not suitable for drier circumstance because it was only tested for wetter circumstances in Canada. But that is not what is said in the Granger-Gray 1989 paper. In fact, the opposite! First, they refer to the data coming from the semi-arid climate zone of Western Canada. Second, if one looks at Figure 2 in that paper we see that the relationship is fitted to G values smaller than 0.6, which means it is most suitable for dry circumstances. → The climate conditions such as semi-arid, wet, and dry depend on the definition of dryness index. As the reviewer mentioned, Granger & Gray (1989) derived G parameter using field data mon-
itored at two stations, Bad Lake and Saskatoon, located in a semi-arid climatic zone of Western Canada. However, according to the data of Centre for Hydrology in University of Saskatchewan (http://www.usask.ca/hydrology/index.php), it can be clearly seen that Saskatoon is a cold region, exhibits classical cold regions hydrology with continuous snow cover from October to April and many lakes and wetlands are present in the central and eastern parts. Furthermore, studies of Anayah & Kaluarachchi (2014) and Xu & Singh (2005) demonstrated that Granger and Gray (1989) model worked well in humid regions and become worse for arid regions. They also suggested that using calibrated parameter values can improve the model performance. Thus, the relative evaporation parameter, G, needs to be upgraded.